# Roles of CgEde1 and CgMca in Development and Virulence of *Colletotrichum gloeosporioides*

**DOI:** 10.3390/ijms25052943

**Published:** 2024-03-03

**Authors:** Dan Wang, Bang An, Hongli Luo, Chaozu He, Qiannan Wang

**Affiliations:** 1Sanya Nanfan Research Institute of Hainan University, School of Tropical Agriculture and Forestry, Hainan University, Sanya 572025, China; 18589344219@163.com (D.W.); anbang@hainanu.edu.cn (B.A.); hlluo@hainanu.edu.cn (H.L.); czhe@hainanu.edu.cn (C.H.); 2Hainan Yazhou Bay Seed Laboratory, Sanya 572025, China

**Keywords:** *Colletotrichum gloeosporioides*, Ede1, metacaspase, conidiation, polar establishment, pathogenicity, insoluble proteins

## Abstract

Anthracnose, induced by *Colletotrichum gloeosporioides*, poses a substantial economic threat to rubber tree yields and various other tropical crops. Ede1, an endocytic scaffolding protein, plays a crucial role in endocytic site initiation and maturation in yeast. Metacaspases, sharing structural similarities with caspase family proteases, are essential for maintaining cell fitness. To enhance our understanding of the growth and virulence of *C. gloeosporioides*, we identified a homologue of Ede1 (CgEde1) in *C. gloeosporioides*. The knockout of *CgEde1* led to impairments in vegetative growth, conidiation, and pathogenicity. Furthermore, we characterized a weakly interacted partner of CgEde1 and CgMca (orthologue of metacaspase). Notably, both the single mutant Δ*CgMca* and the double mutant Δ*CgEde1*/Δ*CgMca* exhibited severe defects in conidiation and germination. Polarity establishment and pathogenicity were also disrupted in these mutants. Moreover, a significantly insoluble protein accumulation was observed in Δ*CgMca* and Δ*CgEde1*/Δ*CgMca* strains. These findings elucidate the mechanism by which CgEde1 and CgMca regulates the growth and pathogenicity of *C. gloeosporioides*. Their regulation involves influencing conidiation, polarity establishment, and maintaining cell fitness, providing valuable insights into the intricate interplay between CgEde1 and CgMca in *C. gloeosporioides*.

## 1. Introduction

Endocytosis is a fundamental cellular process wherein vesicles are formed through the endoplasmic membrane, facilitating the transport of extracellular substances into the cell. This vesicle transport pathway is widespread in eukaryotic cells [1]. It has been reported that endocytosis-related proteins play a very important role in the growth, development, and pathogenicity of filamentous fungi [2], such as Yup1 in *Ustilago maydis* [3], End4 in *Aspergillus nidulans* [4], End3 in *Magnaporthe oryzae* and *Colletotrichum gloeosporioides* [5,6], and Pal1 and Sla1 in *Magnaporthe oryzae* [7]. Ede1 (**E**H **d**omain-containing and **e**ndocytosis protein **1**) primarily functions in the early stage of endocytosis, and its aggregation plays a crucial role in the initiation and maturation of endocytosis sites [8,9]. In *Saccharomyces cerevisiae*, the deletion of *Ede1* does not impact the growth but results in a reduced initiation frequency of endocytosis sites and a decrease in the lifetime of other endocytosis-related proteins, leading to a 35% decrease in endocytosis [10,11,12]. In *Fusarium graminearum*, the loss of FgEde1 leads to slower vegetative growth, decreased spore production, and a significant reduction in pathogenicity. In addition, FgEde1 also participated in the regulation of mycelium endocytosis and autophagy [13]. These findings underscore the crucial role of Ede1 and related proteins in coordinating endocytosis processes.

Apoptosis, a form of programmed cell death, which has been found to be associated with the maintenance of cell homeostasis, regulation of cell growth, differentiation and senescence [14,15]. Caspase family proteases engage in regulating multiple cell behaviors, including apoptosis, that contribute to organism fitness and pathology in metazoan species [16]. Although direct homologs of caspase are not found in a single-celled organism, a group of structural orthologous of caspase termed metacaspases were uncovered in protozoa, plants, and fungi by bioinformatic analysis [17,18,19,20]. Previous functional studies about metacaspases in fungi have demonstrated that fungi usually express 1–2 metacaspase-coding genes [17,21,22,23,24]. The yeast Metacaspase Yca1 is involved in the removal of insoluble protein aggregates and engages in the induction of apoptosis-like cell death under oxidative stress [17,25,26]. Similar functions of metacaspases have been observed in other fungi, including *Podospora anserine* [27,28], *Candida albicans* [22], *Aspergillus fumigatus* [21], *Aspergillus flavus* [29], *Ustilago maydis* [23], and *Magnaporthe oryzae* [24]. Additionally, metacaspases also participate in the regulation of fungal development, energy transformation and secondary metabolism [22,24,30]. These findings highlight the diverse and crucial roles of metacaspases in fungal biology.

The *Colletotrichum* species are a group of notorious pathogenic fungi, which can invade more than 3200 plant species, such as rubber trees, olive fruits, strawberries, etc., and cause anthracnose diseases worldwide [31,32,33,34]. Most *Colletotrichum* species employ a hemibiotrophic and multistage strategy during the infection life cycle. Elucidating the biology of *Colletotrichum* development and infection processes could provide information for controlling strategies against anthracnose diseases. The functions of Ede1 and metacaspase in phytopathogenic fungi remain poorly understood, which requires further investigation. In this study, we first identified a yeast Ede1 ortholog, CgEde1, in *C*. *gloeosporioides*. We succeeded in obtaining a knock-out mutant of *CgEde1*, and found its role in both vegetative growth and pathogenicity of *C*. *gloeosporioides*. Combining pull-down assays of the CgEde1-GFP-expressing strain and mass spectrometry analysis, we identified a potential interacting partner of CgEde1 and CgMca. Subsequent yeast two hybrid assays revealed a weak interaction between the N-terminal of CgEde1 and CgMca. We then constructed the single-mutant ΔCg*Mca* and double-mutant Δ*CgEde1*/Δ*CgMca*, finding that both CgEde1 and CgMca are involved in the regulation of conidiation, infection to the host tissues, and stress tolerance. Furthermore, CgMca were also found to be engaged in keeping cell fitness by eliminating insoluble fractions. Interestingly, the double mutant strain, but not the single mutants, exhibited impairment in polar germination. Collectively, our findings develop the understanding of how CgEde1 and CgMca jointly regulate the growth and pathogenesis of fungal pathogens, and this study may decipher a key role of CgEde1 and CgMca in fungus conidiation and polar establishment.

## 2. Results

### 2.1. Bioinformatic Analysis of CgEde1 and CgMca and Generation of Mutant Strains

The EH domain-containing and endocytosis protein 1 (Ede1) coding gene *CgEde1* and coding gene *CgMca* of the programmed-cell-death-related protein Metacaspase were identified through a BLASTP search of the *C*. *gloeosporioides* genome database, in which the *Saccharomyces cerevisiae* Ede1 (accession: QHB06724.1) and *Saccharomyces cerevisiae* Metacaspase (accession: EDN63533.1) protein sequence were used as queries. The predicted *CgEde1* contains a 3783-bp open reading frame and encodes a 1261-aa protein, while *CgMca* is predicted to contain a 1260-bp open reading frame and encode a 419-aa protein. Phylogenetic analysis revealed that both CgEde1 and CgMca and their respective orthologs from different plant pathogenic fungi are evolutionarily conserved (Appendix A). 

To investigate the physiological roles of CgEde1 and CgMca, the coding genes were knocked out via the split-marker recombination strategy in *C*. *gloeosporioides* (Appendix A). In addition, the knockout mutant Δ*CgEde1* was complemented by introducing a *CgEde1* expression cassette driven by its native promoter, and the complementation mutant was named Res-Δ*CgEde1* (Appendix A).

### 2.2. CgEde1 Is Involved in Vegetative Growth, Conidiation, and Pathogenicity

To assess the significance of CgEde1 in vegetative growth, the growth rate of *C*. *gloeosporioides* strains on PDA and CM media was determined. The results revealed a reduction in the colony growth of Δ*CgEde1* by approximately 16% compared with that of the WT and Res-Δ*CgEde1* (Figure 1A,B). In addition, the deletion of *CgEde1* apparently affected the conidiation of *C*. *gloeosporioides*, with only 61.06% of the spore production observed in the WT strain (Figure 1C). To analyze the role of CgEde1 in the pathogenicity of *C*. *gloeosporioides*, infection tests were conducted on rubber tree leaves. Intact leaves or pre-wounded leaves were inoculated with different strains to monitor the early penetration and late development processes of the pathogen. When droplets of the conidial suspensions were inoculated on the pre-wounded leaves, Δ*CgEde1* caused smaller lesions compared to the WT and Res-Δ*CgEde1*, although all the strains infected leaves with a disease incidence of 100% (Figure 1D). As shown in Figure 1E, although Δ*CgEde1* was capable of penetrating intact leaves, it induced smaller lesions in comparison with the WT and Res-Δ*CgEde1*. These results indicate that CgEde1 is involved in the vegetative growth, conidiation, and pathogenicity of *C*. *gloeosporioides*. 

### 2.3. CgMca Is a Potential Interacting Protein of CgEde1

Given the essential role of CgEde1 in the vegetative growth and pathogenicity of *C*. *gloeosporioides*, and aiming to gain new insights into its underlying mechanism, we genetically tagged CgEde1 with GFP (Appendix A) and observed the subcellular localization of CgEde1. As shown in Figure 2A, CgEde1-GFP patches were uniformly distributed in conidia, and a polarized distribution was observed in vegetative hyphal tips. Furthermore, putative interacting proteins of CgEde1 were identified through anti-GFP antibody magnetic-beads-mediated pulldown and mass spectrometry (Figure 2B). Using this approach, we identified a potential interacting protein of CgEde1, metacaspase. Ede1 is known to be involved in endocytosis, and a previous study showed that mutants over-expressing *Acmcp* (*Acanthamoeba* Type-I metacaspase) had a higher level of endocytosis [35]. We were therefore curious as to whether CgMca interacts with CgEde1 and plays a role in the vegetative or infective stage of *C*. *gloeosporioides*. To perform the point-to-point identification assay of CgMca and CgEde1, we first conducted self-activating tests, and found that CgEde1 has a self-activating function. Therefore, the N-terminal and C-terminal of *CgEde1* were fused with the bait vector pGBKT7, and further point-to-point identification showed that the N-terminal of CgEde1 weakly interacts with CgMca. These results demonstrate that CgMca is a potential interacting protein of CgEde1.

To characterize the expression pattern of *CgEde1* and *CgMca* in more detail, we quantified their transcript accumulation levels in different fungal tissues, including hypha (HY), spore, germ tubes (germinated after 2 h), appressoria on cellophane and rubber tree leaves (in vivo) following inoculation with conidial suspension. The results show that the transcript levels of *CgEde1* were slightly up-regulated in spores in comparison with HY (Appendix A), while the expression levels of *CgMca* were up-regulated in spores, appressoria, and in vivo infection stages in comparison with HY (Appendix A). The results indicated that *CgMca* may play important roles in the virulence of *C*. *gloeosporioides*.

### 2.4. CgMca Is Required for Conidiation, Spore Morphology and Germination

To further explore the potential biological role of CgMca and CgEde1 in *C*. *gloeosporioides*, we first generated the single-mutant Δ*CgMca* and the double-mutant Δ*CgEde1*/Δ*CgMca*. The results from vegetative growth assay showed that deletion of *CgMca* did not affect the colony growth of *C*. *gloeosporioides* on PDA and CM media (Figure 3). To our surprise, compared with Δ*CgEde1*, both Δ*CgMca* and Δ*CgEde1*/Δ*CgMca* strains showed more severe defects in conidiation, with only 1.48% and 0.47% of the WT level (Figure 4A). In addition, we analyzed the conidiation behavior from the germinated hypha on cellophane through microscopy and calculated the number of spores produced by each strain after germinating for 36 h. As shown in Figure 4B,C, in comparison with WT, all three mutants demonstrated a dramatic reduction in spore production. To further determine the impact of *CgEde1* and Δ*CgMca* on conidiogenesis, quantitative real-time-PCR (qRT-PCR) was used to examine the expression pattern of conidiation-related genes in *C*. *gloeosporioides*. As shown in Figure 4D, the expressions of *CgCOS1*, *CgHOX2*, *CgHOX2*, *CgHOX7*, *CgCon6*, *CgCon8*, and *CgCon10* were sharply down-regulated in Δ*CgEde1*, Δ*CgMca*, and Δ*CgEde1*/Δ*CgMca* strains. During these conidiogenetic assays, we also noticed that the loss of *CgMca* apparently affected the morphology of spores. Compared with the WT and Δ*CgEde1*, spores of Δ*CgMca* and Δ*CgEde1*/Δ*CgMca* strains were thinner (Figure 4E–G). Taken together, these results indicate that *CgMca* is essential for conidiation and spore morphology in *C*. *gloeosporioides*.

To analyze the role of *CgEde1* and *CgMca* in conidial germination, we assessed the germination rate of spores from WT and mutant strains. The results indicate that, after gemination for 2 h, 89% and 86.3% of the spores from WT and Δ*CgEde1* germinated successfully, compared to only 2.67% for both Δ*CgMca* and Δ*CgEde1*/Δ*CgMca* strains (Figure 5A,B). Additionally, we observed that almost all the germ tubes germinated from terminals of the spores at 6 h, while 7.17% of spores from Δ*CgEde1*/Δ*CgMca* germinated from the lateral side of the spores, not from either end (Figure 5A,C). These results suggest that CgMca is required for conidial germination in *C*. *gloeosporioides*.

### 2.5. CgMca Is Required for the Virulence of C. gloeosporioides

To analyze the role of CgMca in the virulence of *C*. *gloeosporioides*, we conducted infection tests on rubber tree leaves as described earlier. Regardless of whether pre-wounded or intact leaves inoculated with deletion mutant strains were used, visible lesions caused by Δ*CgEde1*, Δ*CgMca*, and Δ*CgEde1*/Δ*CgMca* were significantly decreased compared to WT (Figure 6A–F). We also noticed that Δ*CgEde1* and Δ*CgMca* caused lesions of similar size, and lesions induced by Δ*CgEde1*/Δ*CgMca* were significantly smaller than those caused by the single mutants. On apple fruit, visible lesions elicited by mutant inoculations were also significantly reduced (Figure 6G,H), consistent with the trend in rubber tree leaves. Overall, these results indicate that *CgMca* were required for the full virulence of *C*. *gloeosporioides*.

### 2.6. Both CgEde1 and CgMca Are Involved in the Development of Invasive Hyphae

To determine why the loss of *CgEde1* and *CgMca* impairs the pathogenicity of *C*. *gloeosporioides*, we tested appressorial formation and the development of invasive hyphae by incubation of *C*. *gloeosporioides* conidia on hydrophobic surfaces and an onion epidermis. After inoculation on artificial hydrophobic surfaces, Δ*CgEde1*, Δ*CgMca*, and Δ*CgEde1*/Δ*CgMca* strains showed similar appressoria formation rates to those of the WT strains (Figure 7A,B). To further validate the reduction in the virulence of all three mutants, we performed a penetration assay using onion epidermis. After incubation for 12 h, 75.33% of the conidia of WT formed invasive hyphae (also named primary hyphae), while the formation rate of the invasive hypha of Δ*CgEde1*, Δ*CgMca*, and Δ*CgEde1*/Δ*CgMca* strains was only 23.33%, 18.00%, and 20.33%, respectively (Figure 7C,D). Although with the extension of inoculation time, most spores of mutant strains penetrated the onion cells at 24 h post-inoculation (hpi), their infection process was significantly slower than that of the WT strain (Figure 7E). These results suggest that the decrease in the virulence of Δ*CgEde1* and Δ*CgMca* is mainly due to their defects in the penetration of plant tissues.

### 2.7. CgEde1 and CgMca Are Involved in Stress Tolerance

To investigate the roles of CgEde1 and CgMca in stress tolerance, the growth rate of *C*. *gloeosporioides* strains on minimum media supplemented with various chemicals was assayed. The results show that the sensitivity of Δ*CgEde1* and Δ*Cgede1*/Δ*CgMca* to H_2_O_2_ and SDS was significantly reduced in comparison to that of the WT and ΔCg*Mca*, and the sensitivity to sorbitol and CFW was slightly increased, whereas the ΔCg*Mca* strain showed a lower sensitivity to sorbitol in comparison with other three strains (Figure 8). These results suggest an impairment in the stress tolerance of the three mutants.

### 2.8. CgMca Plays a Role in the Removal of Protein Aggregates

Previous studies about yeast metacaspase Yca1 revealed its role in keeping protein homeostasis by clearing insoluble protein aggregates [26]. Therefore, we hypothesized that CgMca may function in a similar way. Then, the insoluble fractions in WT and three mutant strains were extracted and detected under SDS-PAGE and subsequently stained with silver. After the removal of cell debris, equal amounts of the supernatant (total cell lysate, TCL) were centrifuged to obtain the insoluble fractions (Figure 9A). As shown in Figure 9B, the Δ*CgMca* and Δ*Cgede1*/Δ*CgMca* mutants showed a significant increase in insoluble aggregates in the cells compared to the WT and Δ*CgEde1* strain. The result made us believe that CgMca is required for the clearance of insoluble protein fractions.

## 3. Discussion

Filamentous fungi regulate the polar growth of mycelia through maintaining a delicate balance between endocytosis and exocytosis [36,37], mainly dependent on the transport of new cell wall and membrane components to the tip of the hypha through vesicle transport [37,38]. Among fungi, the functions of Ede1 are well characterized in the yeast of the model organism. However, although FgEde1 in *Fusarium graminearum* was confirmed to play roles in vegetative growth, virulence and autophagy [13], our understanding of how Ede1 functions in other phytopathogenic fungi, especially in *C*. *gloeosporioides*, remains limited. To address this gap, we first identified an Ede1 homologue CgEde1 in *C*. *gloeosporioides*, and performed a basic phenotypic analysis of Δ*CgEde1*. Consistent with findings in *F*. *graminearum* [13], the knockout of *CgEde1* markedly impaired the colony growth, spore production, and pathogenicity of *C*. *gloeosporioides* (Figure 1).

Given that Ede1 is one of the earliest proteins to reach the new endocytosis site in yeast, and its aggregation means so much to the initiation and maturation of endocytosis sites [8,9], we constructed a CgEde1-GFP-expressing strain to investigate the localization of CgEde1 in *C*. *gloeosporioides*. We found that CgEde1 patches were mainly presented in a polarized distribution in hyphal tips, which indicates its potential role in cell polarity and endocytosis (Figure 2A). However, unexpectedly, unlike Δ*FgEde1* in *F*. *graminearum*, the deletion of *CgEde1* did not impact endocytosis in *C*. *gloeosporioides*, as assessed by staining hypha with FM4-64 [13]. To further explore the function of CgEde1 in *C*. *gloeosporioides*, we screened the putative interacting proteins of CgEde1. By using a pull-down and mass spectrometry analysis, we identified a potential target metacaspase, named CgMca (Figure 2B). Subsequent point-to-point identification revealed that the N-terminal of CgEde1 weakly interacts with CgMca (Figure 2C). Previous studies have shown that the interacting proteins of Ede1 are mainly associated with endocytosis [8]. A series of two-hybrid screens revealed that several proteins interacted with yeast metacaspase YCA1 [39,40]. Among those interacting proteins, SRO77 (YBL106c) codes for a protein homolog to Sro7p and to the Drosophila lethal giant larvae (lgl) tumor suppressor and engages in polarized exocytosis by regulating SNARE function on the plasma membrane [41]. These findings suggest a potential interaction between Ede1 and metacaspase, implicating their participation in the polar growth of fungi.

Metacaspases, sharing structural similarity with caspase family proteases, are widespread in protozoa, plants, and fungi [20]. Similar to yeast, *U*. *maydis* possesses only one metacaspase, Mca1, with a dual role in the intracellular insoluble protein aggregate clearance and the stimulation of apoptosis [23]. Two metacaspases, MoMca1 and MoMca2, were characterized in *Magnaporthe oryzae*, and the study revealed their functions in regulating stress responses and promoting the removal of insoluble aggregates [24]. We bioinformatically identified only one metacaspase, CgMca, in *C*. *gloeosporioides*, and genetically produced the single-mutant Δ*CgMca* and double-mutant Δ*Cgede1*/Δ*CgMca*. Vegetative phenotypic analysis demonstrated that CgMca is dispensable for vegetative colony growth, but is vital for the conidiation and germination of *C*. *gloeosporioides* (Figure 3 and Figure 4), consistent with observations in *Magnaporthe oryzae* [24]. The loss of *CgEde1* caused roughly half the production of spores, and unexpectedly, Δ*CgMca* and the double mutant of Δ*Cgede1*/Δ*CgMca* demonstrated a 98.52% and even 99.53% decrease in conidiation in the liquid medium (Figure 4A). To uncover deeply reasons, we detected the transcriptional levels of several conidiogenesis related genes, and found that most of these genes were significantly down-regulated in all three mutants (Figure 4D). Furthermore, surprisingly, the spore morphology of Δ*CgMca* and Δ*Cgede1*/Δ*CgMca* turned slender compared to WT and Δ*CgEde1* strain, and the germination rate of both of Δ*CgMca* and Δ*Cgede1*/Δ*CgMca* dramatically decreased by 97.3% at 2 hpi (Figure 4E–G and Figure 5A,B). Polar tip growth is widespread in root hairs, pollen tubes, neurons, and the apex of fungal hyphae, regulated by the cytoskeleton, excytosis and endocytosis, the recruitment of cytoplasmic factors, and the delivery of new membrane components to the polarity site [42,43,44]. During germination, spores undergo swelling, and then a polarized initial emerges, a process akin to the budding observed in yeast [45,46]. In *C*. *gloeosporioides*, germinating spores often exhibit growth at both ends of the long axis, leading to bi-directional germination. Our germination assay revealed that WT and two single mutants germinated in a bi-directional pattern, but 7.17% of the Δ*Cgede1*/Δ*CgMca* germinated at the lateral side of spores, indicating aberrant polarity establishment in the double mutant (Figure 5A,C). Since CgEde1-GFP localizes in the hypha tips during germination (Figure 2A), and SRO77, one of the YCA1-interacting proteins in yeast, functions in polarized exocytosis [39,41], it is reasonable to infer that CgEde1 interacts with CgMca and participates in the regulation of polarity establishment during germination.

As a hemibiotrophic fungus, *C*. *gloeosporioides* infect the host via a specialized dome-shaped infection cell known as the appressorium [47]. After maturation and repolarization of the appressorium, an invasive peg emerges and forces through the leaf cuticle. In this study, our investigation indicates that both single mutants showed reduced pathogenicity and the double mutant was even less virulent to host tissues. However, all three mutants retained the capacity to infect intact leaves and induce anthracnose (Figure 6). In order to explore the potential biological role of *CgEde1* and *CgMca* in pathogenicity, we performed appressoria-induced examinations on a hydrophobic surface and onion epidermis. The results suggest that both *CgEde1* and *CgMca* were dispensable for appressorium formation, but is engaged in the early stage of infection to the onion epidermis (Figure 7). During the invasion stage, the immune responses from host tissues, such as ROS burst and thus cause stress for the fungi. Additionally, metacaspases in several fungi have a function in oxidative stress tolerance [23,24]. Therefore, we performed stress tolerance assays, and the data show that, compared to CgMca, CgEde1 may be more important in the responses of these selected stresses (Figure 8). Similar to Δ*Ede1* in *F*. *graminearum*, Δ*CgEde1* demonstrated increased tolerance to oxidative stress [13]. However, in contrast to metacaspases in yeast, *F*. *graminearum* and *Magnaporthe oryzae*, the loss of *CgMca* did not affect the response to oxidative stress in *C*. *gloeosporioides* (Figure 8). Not coincidentally, in *Aspergillus fumigatus*, the CasA/CasB metacaspases also showed no engagement in oxidative stress, as well as some other stresses, such as hyperosmotic stress, salt stress, and heat shock [21]. Considering previous studies indicating the involvement of metacaspases in the clearance of insoluble fractions, we wonder if CgMca functions in a similar way. Just as we expected, the loss of CgMca resulted in the accumulation of insoluble protein fractions in Δ*CgMca* and Δ*Cgede1*/Δ*CgMca*. These findings suggest that both CgEde1 and CgMca are involved in the early invasive stage, and CgMca is crucial for maintaining cell fitness in *C*. *gloeosporioides*.

In summary, our findings suggest that CgEde1 is engaged in the regulation of vegetative growth, conidiation, virulence, and stress tolerance of *C*. *gloeosporioides*, and its weakly interacting protein, CgMca, is involved in the conidiation, pathogenicity, and clearance of insoluble protein aggregates. Moreover, CgEde1 and CgMca appear to collaborate, exerting a certain influence on the polarity establishment during spore germination.

## 4. Materials and Methods

### 4.1. Characterization and Phylogenetic Analysis of CgEde1 and CgMca

To identify EH domain-containing and endocytosis protein 1 (Ede1) and the programmed-cell-death-related protein metacaspase (Mca) in *C*. *gloeosporioides* isolated from *Hevea brasiliensis* (BioSample: SAMN17266943 [https://www.ncbi.nlm.nih.gov/biosample/17266943] (accessed on 9 January 2021)) [48], the amino acid sequences of *S*. *cerevisiae,* Ede1, and *Magnaporthe oryzae,* Metacaspase1 and Metacaspase2, were used to search against the genome of *C*. *gloeosporioides*. One putative Ede1 gene, named *CgEde1*, and one putative metacaspase gene, named *CgMca*, were identified in *C*. *gloeosporioides*. The phylogenetic relationship between CgEde1 and CgMca with orthologs from other filamentous fungi was analyzed through the construction of a maximum-likelihood tree via 1000 bootstrap with MEGA 11 [49].

### 4.2. Generation of Mutant Strains

For single mutants, both *CgEde1* and *CgMca* were knocked out via a split-marker recombination strategy, as shown in Appendix A, according to our previous work [50]. For the deletion of *CgEde1*, single conidia isolations were conducted with six positive transformants (Appendix A), and then eight independent knock-out mutants of *CgEde1* were verified for the gene (Appendix A). Given the similarity in phenotypes among the three randomly selected independent knock-out mutants of *CgEde1*, only one mutant was selected for the gene complementation, and the following analysis. The deletion of *CgMca* was carried out in a similar way to that of *CgEde1* (Appendix A). For the generation of double mutants, the *CgEde1* deletion mutants were used as the recipient strain to conduct the protoplast preparation and transformation to delete *CgMca* (Appendix A). The *CgEde1* knock-out mutant was complemented by introducing the *CgEde1* expression cassette, which comprises the *CgEde1* sequence, together with its native promoter, the terminator of tryptophan synthase of *A*. *nidulans* (TtrpC), and the hygromycin phosphotransferase gene (*HPT*) (Appendix A). The CgEde1-GFP strain was constructed via a homologous recombination strategy, as shown in Appendix A. The primers used are listed in Appendix A.

### 4.3. Vegetative Growth Assay

The vegetative growth of *C*. *gloeosporioides* was examined by culturing the strains on potato dextrose agar medium (PDA, pH6.5) and complete medium (CM, pH 6.5) at 28 °C, and colony morphology and diameter were recorded. Each strain contained three replicates, and all of the experiments were performed twice.

### 4.4. Conidiation Assay

For the conventional conidiation assay, 5 mm diameter disks from different strains growing on the PDA medium were transferred and inoculated into 20 mL liquid complete media (CM) for 2 days. Then, 1 mL spore suspension (10^3^ conidia mL^−1^) from each strain was added to 30 mL liquid CM and cultured at 28 °C with shaking (120 rpm) for 3 days, and the conidia numbers were calculated under a microscope.

For the conidiation assay on cellophane, 20 μL spore droplets (2 × 10^5^ conidia mL^−1^) from each strain were incubated on cellophane and cultured at 28 °C for 36 h, and the conidiation behavior was captured using a Leica DM2000 microscopy (Leica, Mannheim, Germany).

### 4.5. Pathogenicity Assay

For the pathogenicity assay on rubber leaves, 5 μL *C*. *gloeosporioides* conidial suspension (2 × 10^5^ conidia mL^−1^) was drop-inoculated onto rubber tree leaves, either previously wounded or intact [51]. The inoculated leaves were maintained in Petri dishes at 28 °C under natural illumination for 3 days. Subsequently, disease incidence and lesion diameter were recorded. Each treatment comprised three replicates of 10 leaves, and the experiment was repeated three times.

For inoculation with fruits, uniform-sized apple fruits without pre-existing injuries were selected. Then, a wound was created on each fruit at the equator using a sterilized scalpel, and a 5 μL conidia suspension with a concentration of 2 × 10^5^ conidia mL^−1^ was inoculated into the wounds. The inoculated apple fruits were then placed in plastic boxes and kept at 28 °C under 90% relative humidity for 6 days. Disease incidence was calculated with the following formula: (number of fruits with disease symptoms/total number of fruits) × 100%. Each treatment involved three replicates of 5 fruits, and the experiment was repeated twice.

### 4.6. Fluorescence Microscopy

To visualize the subcellular localization of CgEde1 in conidia and hyphae, conidial droplets from CgEde1-GFP expressing strain (5 × 10^5^ conidia/mL) were suspended in YCS media (1 g/L yeast extract, 1 g/L casein, 342 g/L sucrose) on glass slides within a moist chamber at 28 °C for 4 h. Then, the CgEde1 structure in conidia and the tips of germinated hyphae were captured using a laser scanning confocal microscope (LSM880; Zeiss, Oberkochen, Germany) with an excitation of 488 nm through an argon laser and an emission wavelength range of 505–525 nm. All microscopy images were analyzed using ImageJ software (version 1.47 g).

### 4.7. Pull-Down Assay and Mass Spectrometry Analysis

For screening interacting proteins of CgEde1, mycelia from both the GFP control strain and the CgEde1-GFP-expressing strain were harvested and ground into powder with liquid nitrogen. For cell lysis, 5 mL lysing buffer (1 M Tris-HCl (PH = 8), 150 mM NaCl, 1 mM EDTA, 1% Triton-X-100, 1 mM PMSF and 1× Cocktail) were added to each sample. The samples were vortexed for 30 s and then incubated at 4 °C for an additional 30 s. After repeating 3 times, the cell lysates were processed using centrifugation at 6000 rpm for 10 min under 4 °C. Then, the supernatants were transferred into new tubes and subjected to centrifugation at 12,000 rpm for 20 min under 4 °C. The supernatants were collected and incubated with 20 μL anti-GFP magnetic beads (MBL) at 4 °C for 12 h with gentle mixing. The beads were collected using a magnetic rack and washed 3 times with the lysing buffer. The bead-bound protein samples were divided into two parts, one of which was separated on 10% SDS-PAGE gels and stained by silver staining kits (Beyotime, Shanghai, China), and the other part was analyzed by liquid chromatography–mass spectrometry/mass spectrometry (LC-MS/MS), which was performed at Institute of Botany, the Chinese Academy of Sciences.

### 4.8. Yeast Hybrid Assays

For the yeast two hybrids, the full-length cDNA of *CgEde1* was fused with the bait vector pGBKT7. Since CgEde1 have a self-activating function, both the N-terminal and C-terminal sequences of *CgEde1* were used to construct the bait vector pGBKT7, named CgEde1-N and CgEde1-C. CgMca were constructed into pGADT7 and co-introduced with CgEde1-N or CgEde1-C in pairs into the Y2hGold yeast strain. Yeast cells were gradually cultured on SD-Trp-Leu (DDO), SD-Trp-Leu-His (TDO) and SD-Trp-Leu-His-Ade (QDO) plates, respectively, for further growth testing.

### 4.9. Appressorium Formation Assay

For the appressorium formation assay, conidium droplets (5 × 10^5^ conidia mL^−1^) were incubated on a hydrophobic surface, and the appressorium formation behavior was captured after incubation for 12 and 24 h using a Leica DM2000 microscopy (Leica, Mannheim, Germany).

### 4.10. Penetration Ability Assay

For penetration assays, conidia resuspended with ddH_2_O at a concentration of 3 × 10^5^ conidia mL^−1^ were inoculated on onion epidermis plated on water agar plates. After incubation for 12 and 24 h, the infection structures were observed. Each treatment contained three replications, with each replicate comprising at least 200 conidia, and the entire experiment was conducted twice.

### 4.11. Stress Tolerance Assay

The stress tolerance of the strains was evaluated according to our previous research [51]. The chemicals of 10 mmol L^−1^ H_2_O_2_, 1 mol L^−1^ sorbitol, 0.7 mol L^−1^ NaCl, 0.25 mg L^−1^ Congo red, 200 μg L^−1^ CFW, and 0.005% SDS were used to simulate stresses. Each treatment contained three replicates, and all experiments were performed twice.

### 4.12. Quantitative RT-PCR Analysis

For the analysis of the expression profiles of *CgEde1* and *CgMca* in different developmental stages, samples were collected from vegetative growth hypha (HY), spores, germinated tubes (2 h, Gt), in vitro induced appressoria (4 h, 6 h), and lesions on rubber tree leaves that inoculated with conidia for 2 d and 3 d (also called in vivo stage). For the detection of the expression levels of conidiation-related genes in WT and mutant strains, vegetative growth hyphae were collected. Additionally, total RNA isolation, first-strand cDNA synthesis, and qPCR analysis were carried out as previously described [50]. After reverse transcription, all the cDNA samples were diluted to 100 ng/μL for subsequent qPCR analysis. Relative transcription levels were estimated with β_2_-tubulin as the endogenous control and the wild type strain (WT) was used as a reference sample. Each reaction contained three biological replicates. Relative expression levels were estimated using the 2^−ΔΔ*Ct*^ method.

### 4.13. Protein Aggregate Assays

Soluble and insoluble aggregate fractions of total cell proteins of *C*. *gloeosporioides* were extracted as previously described [23,24], with some modifications. Briefly, mycelia from indicated strains of *C*. *gloeosporioides* were harvested and ground in liquid nitrogen. For each strain, a total of 200 mg of ground mycelia was used to performed protein extractions. An equal volume of lysis buffer (0.1% Triton X-100, 50 mM Tris, pH 7.4, 1 mM EDTA, and 1% glycerol supplemented with 5 mM Na_3_VO_4_ and 1 mM phenylmethylsulfonyl fluoride) was added to the samples. Then, the samples were vortexed for 20 min with 1 min on/off cycles, and the resulting suspension lysates were centrifuged at 2000 rpm to remove cell debris. The obtained supernatant was collected and labeled as total cell lysate (TCL). Subsequently, the protein concentration of TCL was determined and normalized for the following analysis. Equal amounts of TCL from different strains were then centrifuged at 12,000 rpm for 15 min at 4 °C to separate the supernatant (soluble fraction) and the pellets (named as insoluble fraction). The insoluble fractions were washed twice with lysis buffer supplemented with 10% Triton X-100, and the final pellets were then re-suspended in 4 M urea. For the analysis of proteins present in TCL and insoluble fractions, the desired amounts of the samples were directly boiled with 5× SDS-PAGE loading dye for 5 min, and were separated by loading on 10% SDS-PAGE gels for subsequent silver staining.

### 4.14. Statistical Analysis

Statistical significance analyses were performed in PASW Statistics 18 (IBM, Armonk, NY, USA). Data with a single variable were analyzed using one-way ANOVA, and mean separations were performed using Duncan’s multiple range test. Differences at *p* < 0.05 were considered significant.

## Figures and Tables

**Figure 1 ijms-25-02943-f001:**
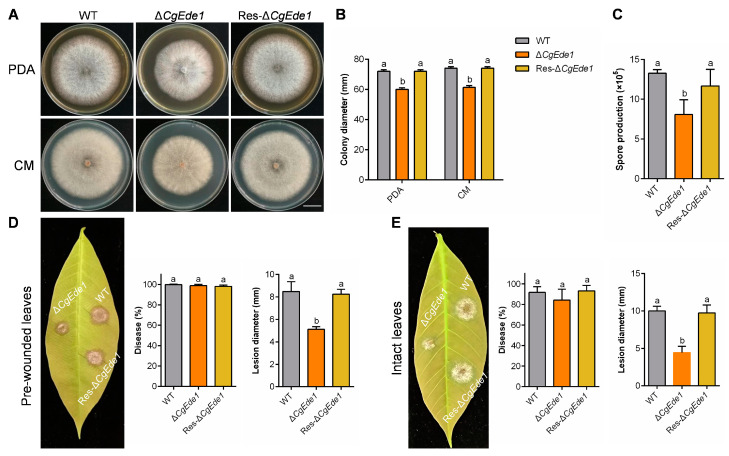
The effects of *CgEde1* deletion on the growth, sporulation, and virulence of *C. gloeosporioides*. (**A**) Colony morphology of WT, ∆*CgEde1*, and Res-∆*CgEde1* incubated on PDA and CM media. Scale bar = 2 cm. (**B**) Colony diameter of the strains on PDA and CM media at 5 d. (**C**) Quantitative analysis of spore production of the strains. (**D**) Disease symptoms, disease incidence and lesion size of **pre-wounded leaves** inoculated with the WT, ∆*CgEde1*, and Res-∆*CgEde1* conidia at 3 dpi. (**E**) Disease symptoms, disease incidence and lesion size of **intact leaves** inoculated with the WT, ∆*CgEde1*, and Res-∆*CgEde1* conidia at 3 dpi. Values are shown as the means ± standard deviations (SD) of three samples. Columns with different letters indicate significant differences (*p* < 0.05).

**Figure 2 ijms-25-02943-f002:**
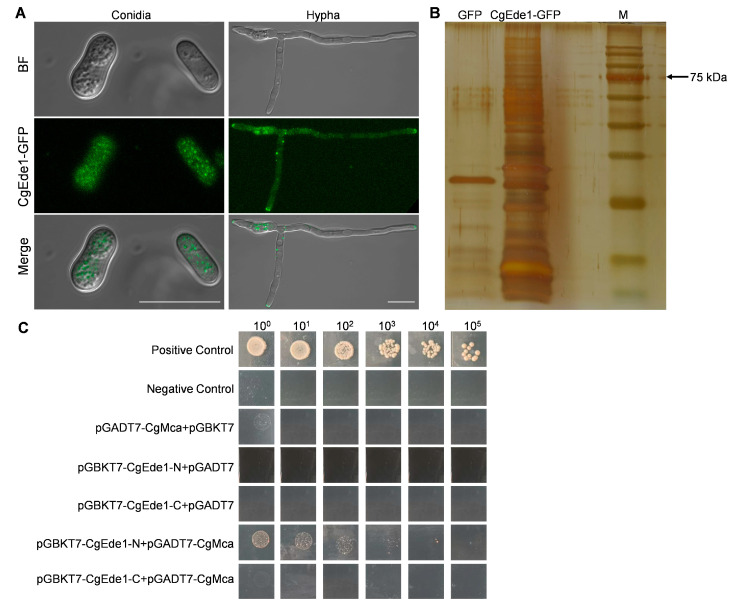
CgEde1 may target the metacaspase protein, CgMca. (**A**) Localization of CgEde1 in the conidia and hyphal tip. Scale bar = 10 μm. (**B**) Separation of purified proteins from GFP expression (control) and CgEde1-GFP strains by SDS-PAGE. The potential interacting protein of CgEde1 were purified using anti-GFP magnetic beads from the CgEde1-GFP strain. The GFP expression strain was used as the control. Proteins in the purified preparations were separated by SDS-PAGE and stained with silver. (**C**) Interaction between CgEde1 and CgMca in yeast. Yeast containing CgEde1 and CgMca grew on SD/−Leu/−Trp/−Ade/−His DO (QDO) plates. pGADT7-T/pGBKT7-Lam were used as negative controls.

**Figure 3 ijms-25-02943-f003:**
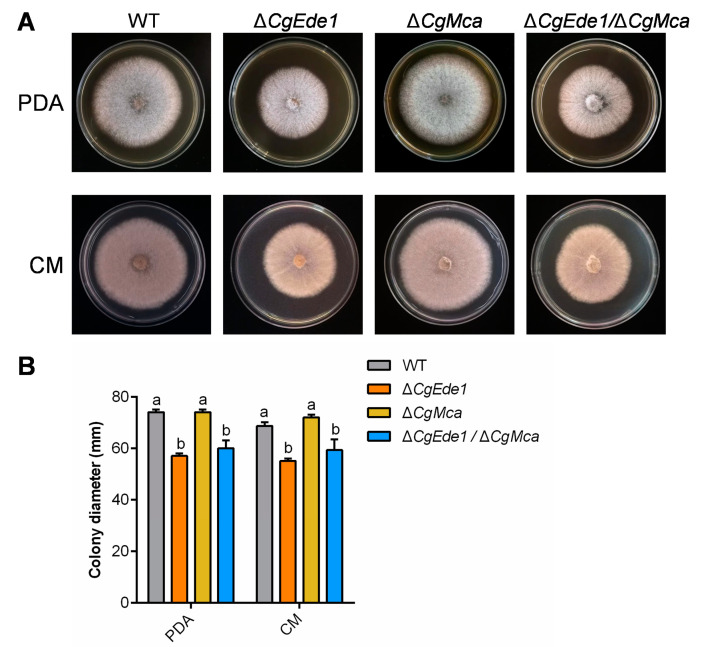
The effects of *CgEde1* and *CgMca* deletion on the growth of *C. gloeosporioides*. (**A**) Colony morphology of WT, ∆*CgEde1*, Δ*CgMca*, and Δ*CgEde1*/Δ*CgMca* incubated on PDA and CM medium. (**B**) Colony diameter of the strains on PDA and CM medium at 5 d. Values are shown as the means ± standard deviations (SD) of three samples. Columns with different letters indicate significant differences (*p* < 0.05).

**Figure 4 ijms-25-02943-f004:**
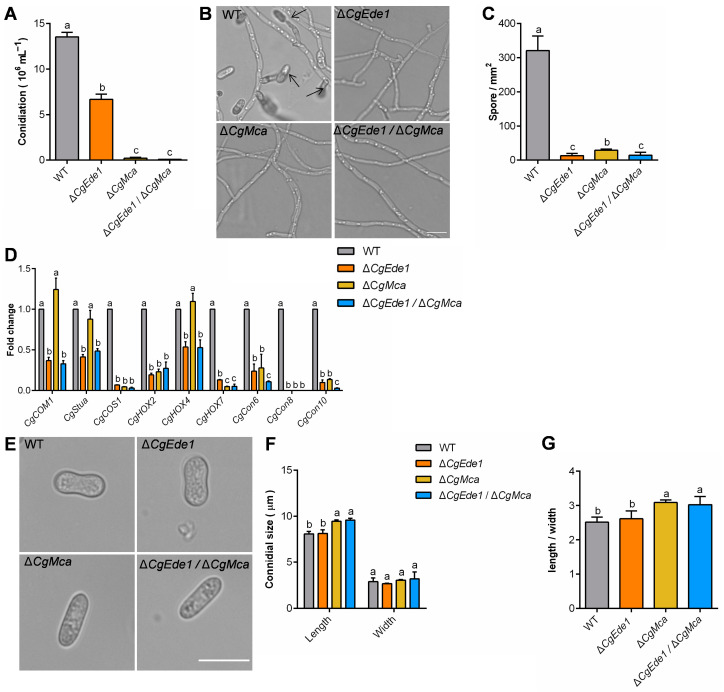
The effects of *CgEde1* and *CgMca* deletion on the conidiation and conidial morphology of *C. gloeosporioides*. (**A**) Conidiation of WT, ∆*CgEde1*, Δ*CgMca,* and Δ*CgEde1*/Δ*CgMca* strains after incubation in liquid complete media for 3 d. (**B**) Sporulation assay conducted by inoculating conidia on the cellophane for 36 h. Scale bar = 10 μm. (**C**) Statistic analysis of the number of spores produced per mm^2^. (**D**) Relative expression level of conidiogenesis-related genes in *C. gloeosporioides*. (**E**) Conidia morphology. Scale bar = 10 μm. (**F**) Conidia size of the strains. (**G**) The ratio of the length to width of the condia. Values are shown as the means ± standard deviations (SD) of three samples. Columns with different letters indicate significant difference (*p* < 0.05).

**Figure 5 ijms-25-02943-f005:**
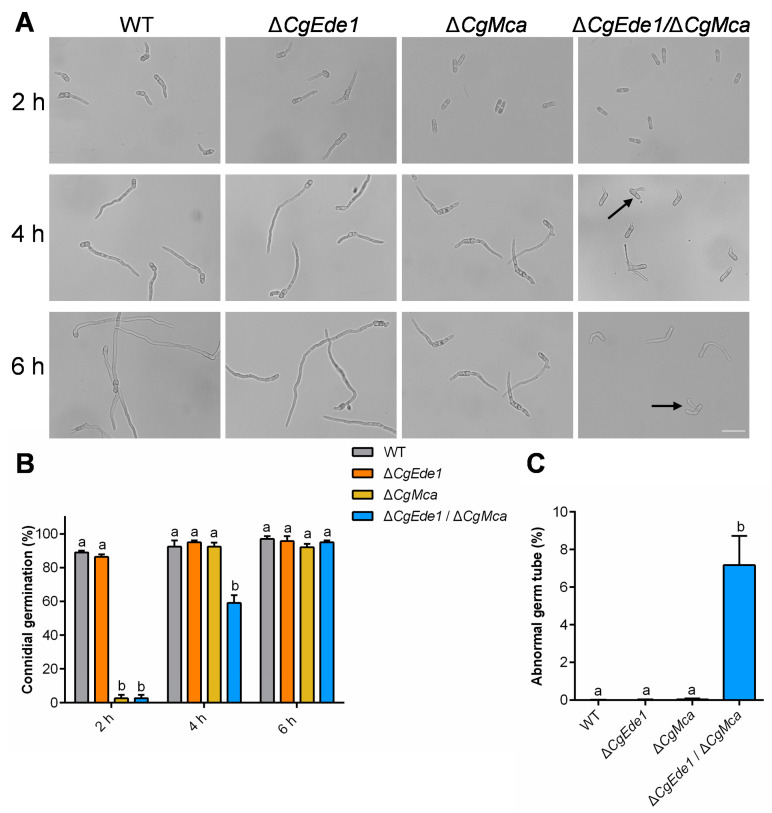
Conidial germination assays of WT, ∆*CgEde1*, Δ*CgMca*, and Δ*CgEde1*/Δ*CgMca* strains. (**A**) Conidial germination of WT, ∆*CgEde1*, Δ*CgMca*, and Δ*CgEde1*/Δ*CgMca* strains after incubation in liquid YCS for indicated time. The black arrow marks the germ tube with an abnormal germination site. Scale bar = 20 μm. (**B**) Statistic analysis of the conidial germination rate. (**C**) Statistic analysis of the abnormal germination rate of spores. Values are shown as the means ± standard deviations (SD) of three samples. Columns with different letters indicate significant difference (*p* < 0.05).

**Figure 6 ijms-25-02943-f006:**
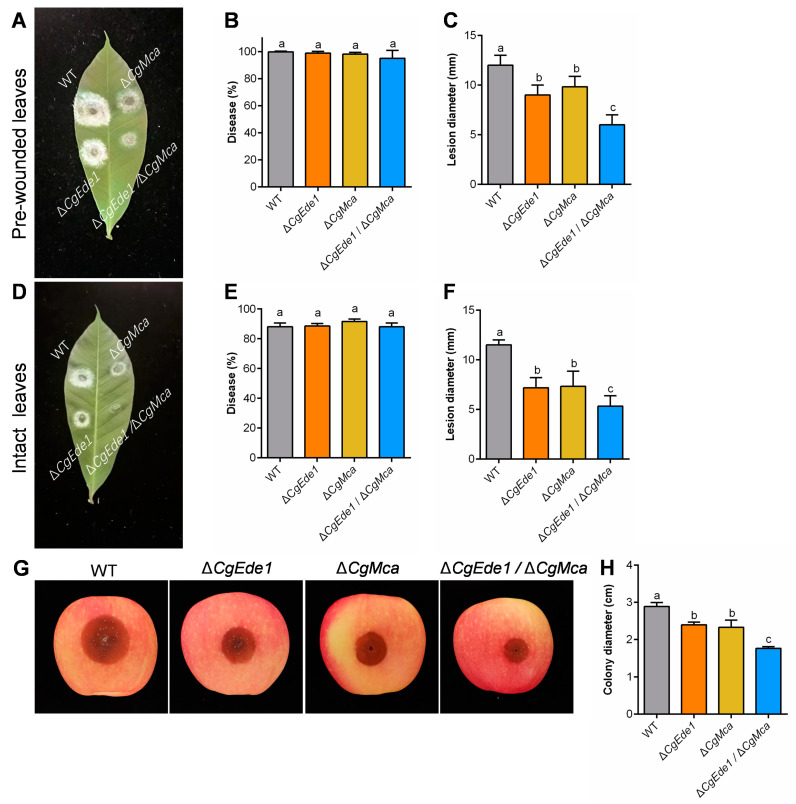
Pathogenicity assays. (**A**) Disease symptoms of WT, ∆*CgEde1*, Δ*CgMca*, and Δ*cgEde1*/Δ*cgMca* strains on the pre-wounded rubber tree leaves at 3 days post-inoculation (dpi). (**B**) Disease incidence of the strains on pre-wounded leaves after 3 dpi. (**C**) Mean lesion diameters on the pre-wounded leaves at 3 dpi. (**D**) Disease symptoms of the strains on the intact rubber tree leaves at 3 dpi. (**E**) Disease incidence of the strains on intact leaves after 3 dpi. (**F**) Mean lesion diameters on the intact leaves at 3 dpi. (**G**) Disease symptoms of the inoculated apple fruits after incubation at 28 °C and 90% relative humidity for 5 d. (**H**) Mean lesion diameters on apple fruit at 5 dpi. Values are shown as the means ± standard deviations (SD) of three groups of samples. Columns with different letters indicate significant differences (*p* < 0.05).

**Figure 7 ijms-25-02943-f007:**
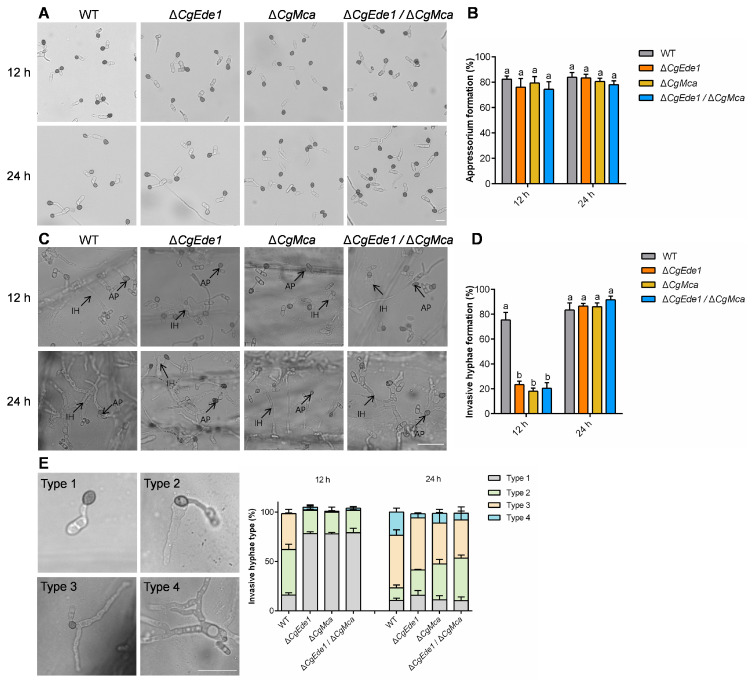
Appressorium formation on hydrophobic surface and onion epidermis. (**A**) Appressorium formation of WT, ∆*CgEde1*, Δ*CgMca*, and Δ*CgEde1*/Δ*CgMca* strains after incubation on hydrophobic surfaces for 12 and 24 h. Scale bar = 10 μm. (**B**) The appressorium formation rate of the strains on hydrophobic polystyrene plates. (**C**) Appressorium formation and penetration of the strains after incubation on onion epidermis for 12 and 24 h. AP: appressorium, IH: invasive hypha. Scale bar = 20 μm. (**D**) The invasive hyphae formation rate of the strains after incubation on onion epidermis for 12 and 24 h. (**E**) The infection severity of the strains after incubation on onion epidermis was observed at 12 and 24 h. Then, the percentages of different types of infectious hyphae in onion cells were counted. Scale bar = 20 μm. Type 1, only with an appressorium; Type 2, only with a single invasive hypha (IH); Type 3, with 1 branch but restricted in one cell; Type 4, with more than 2 branches. Values are shown as the means ± standard deviations (SD) of three groups of samples, and each group of samples comprises at least 100 conidia. Columns with different letters indicate significant differences (*p* < 0.05).

**Figure 8 ijms-25-02943-f008:**
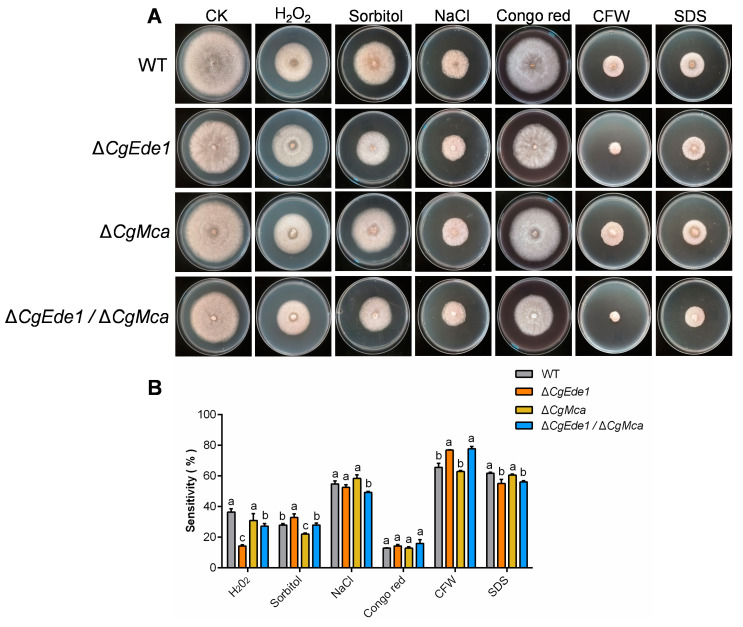
Assays of stress tolerance. (**A**) Colony morphology of WT, ∆*CgEde1*, Δ*CgMca*, and Δ*Cgede1*/Δ*CgMca* strains incubated on minimal medium supplemented with different chemicals for 5 d. CK: control check. (**B**) Quantitative analysis of chemical sensitivity after incubation for 5 d. The sensitivity was calculated by comparing the colony area with that of CK. Values are shown as the means ± standard deviations (SD) of three groups of samples, and each group of samples comprises at least 100 conidia. Columns with different letters indicate significant difference (*p* < 0.05).

**Figure 9 ijms-25-02943-f009:**
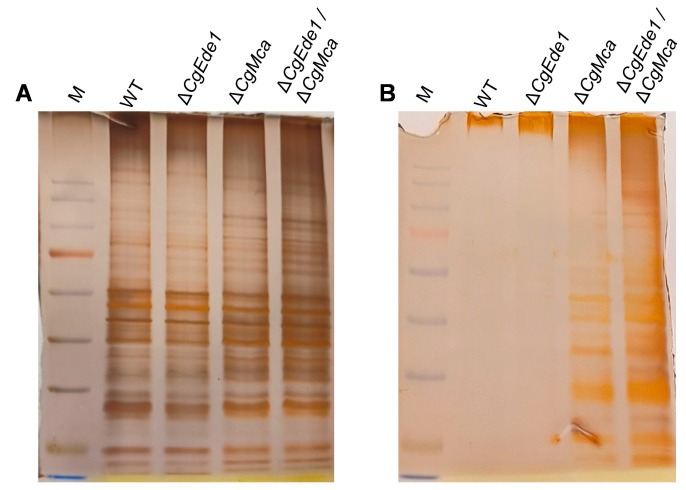
CgMca plays an essential role in clearance of insoluble aggregates. (**A**) Silver-stained 1D PAGE gel of total cell lysate (TCL) of the mycelium of WT, ∆*CgEde1*, Δ*cgMca*, and Δ*Cgede1*/Δ*CgMca* strains. (**B**) Silver staining of insoluble fractions from equal amounts of TCL from different strains.

## Data Availability

The data presented in this study are available on request from the corresponding author.

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
