# Peer review of "Roles of CgEde1 and CgMca in Development and Virulence of Colletotrichum gloeosporioides"

_ijms, 2024, doi:10.3390/ijms25052943_

Round 1
Reviewer 1 Report
Comments and Suggestions for Authors
The article written by Wang et al is interesting. The current MS has written well and experiments were well designed. Although, there are some scope to enhance the quality of the article. Phylogenetic tree should be represented with scale. All the graphs are plotted with looks similar in pattern, it is advisable to use different color (not bright red or green) or at least different pattern filled graph. In my opinion this paper can get accepted after incorporating minor changes, as suggested below.
Minor comments;
Page number 2, Line No. 73: It could be potential interacting partner!
Page number 2, Line No. 73: detections mutants revealed…
Page number 3, Line No. 106: rubber tree leaves were inoculated with different strains of C. gloeosporioides, then why all of them have the similar disease incidence? What do you mean by different strain?
Page number 3, Line No. 115-122: Some of the legend sentence is in bold letter, any significance?
Page number 4, Line No. 142, Figure 2C: Why BD-CgEde1 with AD vector (empty vector) is showing interaction? Did you used inhibitors in YTH assay?
Page number 5, Line No. 172: “expression of most genes tested” What are those genes?
Comments on the Quality of English LanguageMinor typo error!
Author Response
Reviewer 1:
Comments and Suggestions for Authors
1.The article written by Wang et al is interesting. The current MS has written well and experiments were well designed. Although, there are some scope to enhance the quality of the article. Phylogenetic tree should be represented with scale. All the graphs are plotted with looks similar in pattern, it is advisable to use different color (not bright red or green) or at least different pattern filled graph. In my opinion this paper can get accepted after incorporating minor changes, as suggested below.
Answer: We sincerely accept the reviewer’s suggestion. We are sorry for our carelessness, and have added the scales for the phylogenetic tree in revised Figure S1. In addition, all the bar graphs were re-plotted with new color scheme in the revised figures to present a better visual effect.
- Minor comments;
(1) Page number 2, Line No. 73: It could be potential interacting partner!
Answer: We sincerely accept the reviewer’s advice. The phrase “potential interacting protein” in line 73 has been changed to “potential interacting partner”.
(2) Page number 2, Line No. 73: detections mutants revealed…
Answer: We are sorry for the unclear description. We have changed “Subsequent detections revealed” in line 74 to “Subsequent yeast two hybrid assay revealed”.
(3) Page number 3, Line No. 106: rubber tree leaves were inoculated with different strains of C. gloeosporioides, then why all of them have the similar disease incidence? What do you mean by different strain?
Answer: In order to ensure the same inoculation environment for WT, mutant strain ΔCgEde1 and recombination strain Res-ΔCgEde1 as much as possible, we inoculated those three different strains in different locations on the same leaf. Similar disease incidence suggests that all of WT, mutant strain ΔCgEde1 and recombination strain Res-ΔCgEde1 can invade into rubber trees. “Different strains” here represents WT, mutant strain ΔCgEde1 and recombination strain Res-ΔCgEde1.
(4) Page number 3, Line No. 115-122: Some of the legend sentence is in bold letter, any significance?
Answer: “pre-wounded leaves” in line 118-119 and “intact leaves” in line 120 are bolded to make them more visible. In this way, the readers can quickly discover that the leaves used in the Figure 1D and 1E are treated differently.
(5) Page number 4, Line No. 142, Figure 2C: Why BD-CgEde1 with AD vector (empty vector) is showing interaction? Did you used inhibitors in YTH assay?
Answer: In our YTH assay, we found that CgEde1 has self-activating function (the third row in Figure 2C). So, in order to show the self-activating results about CgEde1, the images presented in the third row (pGBKT7-CgEde1+pGADT7) and fifth row (pGBKT7-CgEde1+pGADT7-CgMca) in Figure 2C were selected from the plates without inhibitors. In order to avoid causing misunderstanding again, we have modified the Figure 2C by deleting the images in the third row and fifth row. And we have changed “and the result showed that CgEde1 has self-activating function (Figure 2C)” in line 137 to “and found that CgEde1 has self-activating function (data not shown)”.
(6) Page number 5, Line No. 172: “expression of most genes tested” What are those genes?
Answer: We sincerely accept the reviewer’s advice. We have changed “the expression of most genes tested were sharply down-regulated” in line 172 to “the expression of CgCOS1, CgHOX2, CgHOX2, CgHOX7, CgCon6, CgCon8 and CgCon10 were sharply down-regulated”.

Reviewer 2 Report
Comments and Suggestions for Authors
Please find the comments in the PDF file.

Author Response
Reviewer:
1- INTRODUCTION presents sufficient information on the topic and covers all aspects of the study, but I would like to see a couple of sentences emphasizing the value of this avenue of work. For example, this work may decipher a key component of a common key player in fungus growth and biogenesis.
Answer: We sincerely accept the reviewer’s suggestion. We have added “and this work may decipher a key role of CgEde1 and CgMca in fungus conidiation and polar establishment.” In line 82.
2- MATERIALS & METHODS: I would like to see some specifications that may affect fungal growth in vitro.
(1)Please reference the CM media, is it cm minimal?
Answer: CM is referred to complete medium. We have annotated it in line 417.
(2)Please add the pH values of PDA and CM used in this work.
Answer: We sincerely accept the reviewer’s advice. “potato dextrose agar medium (PDA) and complete medium (CM)” in line 417 has been changed to “potato dextrose agar medium (PDA, pH6.5) and complete medium (CM, pH6.5)”.
(3)I believe that inoculating a hempbiotrophic fungus on detached leaves will not reflect its parasitic capacity on this leaf, since plant defences will decline after detaching the leaves. Therefore, I suggest running the same experiment on attached leaves on young plants to get better clue on gene expression. Inoculating detached leaves will mainly represent the necrotrophic behavior of the fungus.
Answer: We sincerely accept the reviewer’s suggestion. That's very good advice. It is indeed the most intuitive by inoculating on attached leaves. We have been studying on the interaction between rubber tree and Colletotrichum gloeosporioides for nearly 10 years. In our previous work, we found that the differences in pathogenicity between various C. gloeosporioides strains could also be demonstrated by inoculation of detached rubber tree leaves. In addition, our inoculation assay was repeated at least 3 times, and each treatment comprised three replicates of 10 leaves.
(4)Inoculating two distant spots on the same leaf may trigger cross talks at the defense pathways and one of which may impair the other, which will give biased result. I recommend inoculating each pathogen at different leaf, different plant.
Answer: We sincerely accept the reviewer’s advice. In our previous work, for the inoculation of C. gloeosporioides on rubber tree leaves, we have tried to inoculate different strains in different parts on the same leaf, and also inoculated each pathogen strain at different leaf, and found that both inoculation methods can truly reflect the differences in the virulence of different C. gloeosporioides strains. Corresponding reference “[51]” was added in line 432 in the subsection of 4.5 pathogenicity assay.
- Liu, S.K.; Wang, Q.N.; Liu, N.; Luo, H.L.; He, C.Z.; An, B. The histone deacetylase hos2 controls pathogenicity through regulation of melanin biosynthesis and appressorium formation in colletotrichum gloeosporioides. Phytopathology Research 2022, 4, 1– DOI:10.1186/S42483-022-00126-0
(5)The inoculum potential used for the pathogenicity assays is high, which will now allow you to catch the fine differences between isolates. Please provide solid evidence with reference for inoculum conc., or go down to 103.
Answer: In our previous work, for the inoculation of wild type C. gloeosporioides on rubber tree leaves, we have tried the inoculum conc. of 103, and the found that the incidence rate of WT strain is not very stable. And we dropped only 5 uL conidia suspension (2 × 105 conidia mL-1) on each inoculation site, which means about 1000 conidia. The volume of conidia suspension was supplemented in Line 430 and the corresponding reference [51] was added in line 432 in the subsection of 4.5 pathogenicity assay.
(6)What is YCS media?
Answer: We are sorry for the unclear description. YCS stands for Yeast extract, Casein and Sucrose. We have changed “YCS media” in Line 447 to “YCS media (1 g/L yeast extract, 1 g/L casein, 342 g/L sucrose)”.
(7)For qPCR:
(i) please add primer’s location/Table for the target and reference genes.
Answer: We sincerely accept the reviewer’s suggestion. We have added the primer’s location for the genes used in qPCR analysis in revised supplemental material.
(ii) How were data presented? ddCT?
Answer: Yes, the data were presented by the ddCT method. We are sorry for the unclear description. We have added “Relative expression levels were estimated using the 2-ΔΔCt method.” In Line501.
(iii) Please specify the amounts or dilutions used from RNA/cDNA.
Answer: We sincerely accept the reviewer’s advice. We have changed “After reverse transcription and qPCR” in Line 498 to “After reverse transcription, all the cDNA samples were diluted to 100 ng/μL for subsequent qPCR analysis.”
3- RESULTS:
1. Fig.1: even though you see some significance in growth reduction between wt and delEde, this variation is not clearly pronounced on the fungal colony grown on PDA. The sporulation percentage shows more of a significant variation between wt and mutant line.
Answer: We sincerely accept the reviewer’s suggestion. We have readjusted the images about plates in Figure 1A, and the scale bar was added to better present the difference in colony diameter of various strains.
2. Fig, 4: do you have a speculation for the signaling cascade/roles of the different conidiogenious related genes in Colletotrichum spp. I wonder if Ed1 and Mca crosstalk or affet each other via downstream gene.
Answer: This is a very interesting and thought-provoking perspective. We sincerely accept the reviewer’s advice. In filamentous fungi, conidiogenesis requires the temporal and spatial control of cell differentiation, which is a process under polygenic control. CgEde1 and CgMca may affect each other via downstream genes, which need further exploration. And in our future work, we would conduct deeper analysis about the formation of conidia in C. gloeosporioides.
Reviewer 3 Report
Comments and Suggestions for Authors
The manuscript titled "Roles of CgEde1 and
CgMca in Development and Virulence of Colletotrichum gloeosporioides" presents a thorough investigation into the
mechanisms governing the growth and pathogenicity of C. gloeosporioides. The
authors demonstrate clarity and coherence throughout the manuscript, ensuring
that both the results and discussions are accessible to readers. Experimental
rigor is evident in the use of knockout mutants and protein interaction
analyses, enhancing the reliability of the findings. The identification of
CgEde1 and its interaction with CgMca provides valuable insights into fungal
virulence mechanisms. The observed impairments in vegetative growth,
conidiation, and pathogenicity contribute significantly to our understanding of
fungal pathogenesis. However, an expanded introduction providing greater
context on the impact of anthracnose would enhance the manuscript's relevance.
Overall, the manuscript represents a solid contribution to fungal biology
research and is suitable for publication in the follow journal. Specific comments are presents in the PDF file.

Author Response
Reviewer 2:
Comments and Suggestions for Authors
- I would suggest including a little more information on the genus Colletotrichum, to emphasise the importance of studying CgEde1 and CgMca in the development and virulence of Colletotrichum. For example, in Spain and Italy, where the disease is endemic, olive anthracnose is considered as the most damaging disease of olive fruit worldwide. It causes a substantial deterioration of oil quality, the severity of which depends on the proportion of infected fruits, the virulence and the species of Colletotrichum the disease and the olive cultivar. Another example is the severity of anthracnose on strawberries. In the work of Chung et al. 2020, it is shown how the diversity and virulence of different strains belonging to the same species can vary the severity of the disease.
Here are some references that can be used to add completeness to the introduction
Riolo, M., Pane, A., Santilli, E., Moricca, S., Cacciola, S.O. Susceptibility of Italian olive cultivars to various Colletotrichum species associated with fruit anthracnose. Plant Pathology (2023), 72(2), 255–267. https://doi.org/10.1111/ppa.13652
Moral, J., Xaviér, C.J., Viruega, J.R., Roca, L.F., Caballero, J. & Trapero, A. (2017) Variability in susceptibility to anthracnose in the world collection of olive cultivars of Cordoba (Spain). Frontiers in Plant Science, 8, 1892.
Chung, PC., Wu, HY., Wang, YW. et al. Diversity and pathogenicity of Colletotrichum species causing strawberry anthracnose in Taiwan and description of a new species, Colletotrichum miaoliense sp. nov.. Sci Rep 10, 14664 (2020). https://doi.org/10.1038/s41598-020-70878-2
Answer: We sincerely accept the reviewer’s suggestion. We have changed “which can invade more than 3200 plant species and cause anthracnose diseases worldwide [31,32]” to “which can invade more than 3200 plant species, such as rubber trees, olive fruits, strawberries, and so on, and cause anthracnose diseases worldwide [31-34]”. We also added corresponding references as below:
- Riolo, M., Pane, A., Santilli, E., Moricca, S., Cacciola, S.O. Susceptibility of Italian olive cultivars to various Colletotrichum species associated with fruit anthracnose. Plant Pathol. 2023, 72, 255–267.DOI: 10.1111/ppa.13652
- Chung, P. C., Wu, H. Y., Wang, Y. W., Ariyawansa, H. A., Hu, H. P., Hung, T. H., Tzean, S. S., & Chung, C. L. Diversity and pathogenicity of Colletotrichum species causing strawberry anthracnose in Taiwan and description of a new species, Colletotrichum miaoliense sp. nov. Sci. Rep. 2020, 10, 14664. DOI: 10.1038/s41598-020-70878-2
- I would suggest including the references of the isolate of Colletotrichum gloeosporioides used in this study.
Answer: We sincerely accept the reviewer’s advice, and related reference have been added in 4.1 as below:
- Wang, Q., An, B., Hou, X., Guo, Y., Luo, H., & He, C. Dicer-like Proteins Regulate the Growth, Conidiation, and Pathogenicity of Colletotrichum gloeosporioides from Hevea brasiliensis. Front. Microbiol. 2018, 8, 2621. DOI: 10.3389/fmicb.2017.02621
Reviewer 4 Report
Comments and Suggestions for Authors
Summary
The manuscript provides a comprehensive investigation into the functional characterization of Ede1 and Metacaspase (CgEde1 and CgMca) in the filamentous fungus C. gloeosporioides. The study covers various aspects, including molecular characterization, phylogenetic analysis, mutant strain generation, phenotypic analysis, subcellular localization, interaction studies, and stress tolerance assays.
The study aims to elucidate the roles of CgEde1 and CgMca in C. gloeosporioides, contributing to a better understanding of the molecular mechanisms underlying vegetative growth, conidiation, pathogenicity, and stress tolerance.
The methods employed, such as mutant strain generation, phenotypic analysis, fluorescence microscopy, pull-down assays, and stress tolerance assays, are well-described and supported by appropriate controls.
The results are presented clearly, and the findings related to the roles of CgEde1 and CgMca in different developmental stages, stress responses, and pathogenicity are well-supported by experimental evidence.
The manuscript is a valuable contribution to the understanding of fungal biology, specifically focusing on the roles of CgEde1 and CgMca in C. gloeosporioides. The combination of molecular, genetic, and phenotypic analyses provides a robust foundation for the reported findings. Addressing the minor suggestions would further strengthen the manuscript.
In general, the manuscript is exceptionally well-written and appears suitable for publication with only minor revisions. Congratulations on the outstanding work.
Minor comments
Consider breaking down the "Materials and Methods" section into more subsections for improved readability, as it covers diverse experimental approaches.
While the manuscript is generally well-written, a final proofread for minor grammatical and language improvements could enhance the overall quality.
Author Response
Reviewer 3:
Comments and Suggestions for Authors
Minor comments
- Consider breaking down the "Materials and Methods" section into more subsections for improved readability, as it covers diverse experimental approaches.
Answer: We sincerely accept the reviewer’s advice. To improve readability, we have break both “4.3 vegetative growth and conidiation assay” and “4.8 appressorium formation and penetration ability assay” into two subsections.
- While the manuscript is generally well-written, a final proofread for minor grammatical and language improvements could enhance the overall quality.
Answer: We sincerely accept the reviewer’s suggestion. We have checked the whole manuscript to correct various grammatical errors.